# Longitudinal D-Dimer Trajectories and the Risk of Mortality in Abdominal Trauma Patients: A Group-Based Trajectory Modeling Analysis

**DOI:** 10.3390/jcm12031091

**Published:** 2023-01-30

**Authors:** Chuanrui Sun, Fengchan Xi, Jiang Li, Wenkui Yu, Xiling Wang

**Affiliations:** 1Key Laboratory of Public Health Safety, Ministry of Education, School of Public Health, Fudan University, Xuhui District, Shanghai 200231, China; 2Research Institute of General Surgery, Affiliated Jinling Hospital, Medical School of Nanjing University, Nanjing 210093, China; 3Department of Intensive Care Unit, Affiliated Drum Tower Hospital, Medical School of Nanjing University, Nanjing 210008, China

**Keywords:** abdominal trauma, D-dimer, group-based trajectory modeling, low-molecular-weight heparin

## Abstract

This study aimed to identify the long-term D-dimer trajectory patterns and their associations with in-hospital all-cause mortality in abdominal trauma patients. This is a retrospective cohort study of general adult abdominal trauma patients admitted to Jinling Hospital (Nanjing, China) between January 2010 and April 2020. Group-based trajectory modeling was applied to model D-dimer trajectories over the first 50 days post-trauma. A multivariable logistic regression was performed to estimate the associations between D-dimer trajectories and in-hospital all-cause mortality. A total of 309 patients were included. We identified four distinct D-dimer trajectories: group 1 (57.61%; “stable low”), group 2 (28.16%; “moderate-decline”), group 3 (8.41%; “high-rapid decline”), and group 4 (5.83%; “high-gradual decline”). The SOFA score (*p* = 0.005) and ISS (*p* = 0.001) were statistically higher in groups 3 and 4 than in groups 1 and 2. The LMWH and UFH did not differ between groups 3 and 4. Compared with the patients in group 1, only the patients in group 4 were at a higher risk of in-hospital all-cause mortality (OR = 6.94, 95% CI: 1.20–40.25). The long-term D-dimer trajectories post-trauma were heterogeneous and associated with mortality. An initially high and slowly-resolved D-dimer might function as the marker of disease deterioration, and specific interventions are needed.

## 1. Introduction

Traumatic injury is a major public health burden worldwide, resulting in 4.4 million deaths every year [1,2]. Among the different injured regions, abdominal trauma accounts for about 30% of trauma-related injuries [3]. With injuries to the abdominal solid organs, patients reach hypercoagulable states within 12 h of injury [4]. Major American and European guidelines recommend early mechanical or pharmacologic thromboprophylaxis for trauma patients as they are at a high risk of venous thromboembolism (VTE) [5,6,7]. Early mechanical prophylaxis is suggested for patients with a bleeding risk, while pharmacologic thromboprophylaxis, such as low-molecular-weight heparin (LMWH) and unfractionated heparin (UFH), is recommended for trauma patients once bleeding has stabilized.

For the risk assessment of VTE, D-dimer has been widely used as a laboratory biomarker in clinical settings [8]. D-dimer is the smallest degradation product of cross-linked fibrin, and an increase of D-dimer indicates the activation of coagulation and fibrinolysis [9]. The association between elevated D-dimer levels and the mortality of trauma patients has been well documented [10,11]. Yuan et al. showed that patients with higher D-dimer within six hours post-trauma increased their risk of 30-day mortality by 1.8 (95% CI: 1.57–2.06)-fold [12]. Nakae et al. demonstrated that higher D-dimer measurements (10 mg/L increments) at 1 h, 3 h, 6 h, and 12 h post-trauma all contributed to a higher risk of mortality [13]. However, these studies were limited to D-dimer measurement at a single time point or the acute phase after trauma. An elevated single-time D-dimer measurement can result from various factors, and its false positive rate of thromboembolic disease is high [14,15]. D-dimer trajectories, which have not been investigated so far, can provide more information about the heterogenous longitudinal response to clinical treatments in addition to changed trauma severity [16] and complications [17]. The patterns of dynamic D-dimer can also reflect the pathophysiology of disease progression. Tracking the trajectory patterns of D-dimer during trauma hospitalization contributes to a stratification of trauma patients at risk of mortality and provides a new perspective on treatment strategies. Moreover, Naymagon et al. showed that there were two patterns of quadratic trajectories of D-dimer for critically ill patients, indicating the D-dimer trajectory post-trauma might follow a heterogenous nonlinear temporal profile [18].

In this study, we hypothesize that patients with abdominal trauma follow distinct D-dimer trajectories, which are associated with different risks of mortality. To investigate this hypothesis, we specifically address three aims: (1) to identify D-dimer trajectories in abdominal trauma patients using a group-based trajectory modeling (GBTM) approach; (2) to assess the association of D-dimer trajectories with all-cause mortality risk; and (3) to optimize the interventions for patients in high-risk groups to improve the survival rates.

## 2. Materials and Methods

### 2.1. Study Population

Patients with abdominal trauma admitted into Jinling Hospital, which is affiliated with the Medical School of Nanjing University, during the period January 2010–April 2020 were screened for eligibility. Abdominal trauma was defined as a blunt or penetrating injury in the abdominal cavity. The abdominal cavity is between the thorax cephalad and the pelvis caudad, whose upper limit extends from a horizontal plane passing through the base of the xiphoid process and the spinous process of the 12th dorsal vertebra. The lower limit of the abdominal cavity arises from the symphysis pubis, extends along the entire inguinal arc and iliac crest, and ends at the spinous process of the 5th lumbar vertebra [19]. Patients aged ≤18 years of age, pregnant women, and cancer patients were excluded from our analysis as the target population was general adult trauma patients. Re-admitted patients after recovery from primary trauma and patients with trauma-to-hospitalization intervals longer than two weeks were excluded to capture the dynamics of D-dimer in the acute phase. Patients were classified as “re-admitted after recovery” if there was a “recovery” or “re-admission” in their medical records of admission. The trauma-to-hospitalization interval was defined as the time interval between the traumatic injury and admission to our trauma center. We further excluded patients with D-dimer measured at fewer than three days to model the nonlinear longitudinal trajectory. This study was approved by the Human Ethics Committee of the Jinling Hospital (IRB# 2021NZKY-045-01) and the Institutional Ethics Review Board of the School of Public Health of Fudan University (IRB# 2021-11-0930). The informed consent was waived because this study was retrospective based on electronic medical records.

### 2.2. Data Collection

The demographics (i.e., age, gender, and body mass index (BMI)), severity of trauma, complications, length of hospital stay, longitudinal D-dimer measurements, clinical treatments, health outcomes of either discharge/death, and the corresponding dates were extracted from the electronic medical records. The severity of the trauma was evaluated by the Sequential Organ Failure Assessment (SOFA) score, Abbreviated Injury Scale (AIS), Injury Severity Score (ISS), and the probability of survival (Ps) obtained by the Trauma and Injury Severity Score (TRISS) when admitted. The TRISS Ps was computed based on a logarithmic regression equation: survival probability = 1/(1 +$\;e^{-b}$), where b = −0.4499 + 0.8085 × RTS − 0.0835 × ISS − 1.7430 × AgeIndex for a blunt trauma or b = −2.5355 + 0.9934 × RTS − 0.0651 × ISS − 1.1360 × AgeIndex for a penetrating trauma. The value of the AgeIndex was 0 with an age < 55 and was 1 with an age ≥ 55 [20]. There were missing data for the AISs (8.74%) and ISSs (8.74%). The AIS scores were imputed by the k-nearest neighbors (KNN) data imputation technique, and the ISS was calculated from the AIS (see Missing data and Appendix A). A complication of sepsis was diagnosed based on Sepsis-3 [21]. Intra-abdominal infection was confirmed according to the abdominal symptoms, increased inflammatory biochemical markers, and computerized X-ray tomography [22]. VTE was diagnosed by duplex ultrasonography or computed tomography [23]. Renal dysfunction was defined as the two-fold increase in serum creatinine and blood urea nitrogen or anuria for >6 h [24]. Liver dysfunction was defined as serum bilirubin ≥ 2.0 mg/dL [25].

The D-dimer test was performed via an immunoturbidimetric assay using the INNOVANCE D-Dimer assay (Siemens Healthcare Diagnostics Products GmbH, Marburg, Germany) on the Sysmex CS5100 coagulation analyzer (Sysmex Chemical Medicine, Kobe, Japan) in the hospital laboratory. In our center, there were two norms of D-dimer testing: (1) in principle, the D-dimer was measured daily during hospitalization for trauma patients, and (2) a D-dimer test would be stopped if patients had an Acute Physiology and Chronic Health Evaluation II (APACHE II) score < 6, a SOFA < 2, resolved disease, or a recovery of organ dysfunction. We split the D-dimer measurements into 1-day blocks and took the daily maximum value if there were multiple measurements on the same calendar day. The reason we chose the daily maximum D-dimer was because it emphasized the pathological states in one block rather than the minimum value [26]. A maximum of 50 days after trauma was considered for trajectory analysis.

Clinical treatments included UFH, LMWH, blood transfusion, blood products of plasma, blood products of cryoprecipitate, and surgery. UFH and LMWH were the top two commonly used anticoagulation agents in our database. Blood transfusion and surgery were potential factors affecting coagulation progression. The patients were treated with UFH 4100–10,000 IU for thromboprophylaxis or treatment. Patients at a high risk of VTE were treated with LMWH 4100–5000 IU each time for at least five days. The timing of initiation and times of the UFH, LMWH, and blood transfusion depended on the patients’ clinical statuses. The primary outcome was in-hospital all-cause mortality.

### 2.3. Statistical Analysis

We employed GBTM with a censored normal distribution (0–40 mg/L) to identify the temporal patterns of D-dimer measurements in the first 50 days post-trauma. GBTM is a semi-parametric finite-mixture model to identify homogenous subgroups following a similar progression over time [27]. We used a two-stage model selection process suggested by Nagin to determine the optimal number and shape order of the trajectories [28]. In the first stage, we used the Bayesian information criterion (BIC), clinical plausibility, and parsimony of models, to determine the optimal number of trajectory groups from one group to six groups. Considering the flexibility of quadratic polynomials, we specified all groups as quadratic trajectories in this stage. After selecting the best-fitting number of groups, we determined the shape order using the following four statistical properties: (1) the least BIC; (2) entropy ≥ 0.9; (3) proportions of individuals classified in each group ≥ 5%; and (4) the average posterior probability of assignment (APPA) ≥ 0.7. Each group was iteratively fitted with quadratic, cubic, and quartic polynomials [29]. Additional information about the model selection and evaluation can be found in the Appendix A (Model development, Appendix A).

The patients were assigned to the group with the highest posterior probability of group membership (PPGM) based on the best-selected model. The clinical characteristics, D-dimer measurements, and outcomes were compared among the trajectory groups. A logistic regression model was used to assess the association between the D-dimer trajectories and in-hospital all-cause mortality. Baseline variables with *p*-values of < 0.1 in the univariate analysis and minimum potential collinearity were selected for a multivariable analysis. A potential collinearity was defined as a maximum variance inflation factor (VIF) ≥ 10 or correlation coefficients > 0.5 with statistical significance. We further tested the sensitivity of the results after changing the covariate of the AIS of the abdomen into ISS, as the correlation between these two variables was 0.52. The prediction performances of the D-dimer trajectory model and the TRISS model for in-hospital all-cause mortality were compared using the area under the receiver-operating-characteristics curve (AUC).

The continuous variables were summarized as median and interquartile ranges (IQR), and the categorical variables as frequencies and proportions. All the continuous variables were skewed according to the Shapiro–Wilk test and were compared using a Kruskal–Wallis test among the groups, while the categorical variables were compared using a chi-square test or Fisher exact test as appropriate. Pairwise comparisons were adjusted by the Benjamini and Hochberg method. A Spearman’s rank correlation test was used to assess the correlations between the continuous variables. A two-tailed *p* value < 0.05 was considered statistically significant. The analyses were performed using the SAS version 14 software (for GBTM analysis) and R version 4.1.3 software (for other analyses).

## 3. Results

### 3.1. Patient Characteristics

Of 913 patients with abdominal trauma, we excluded 227 patients with the D-dimer measured at fewer than three days, 197 re-admitted patients after recovery from primary trauma, 136 patients with trauma-to-hospitalization intervals longer than two weeks, and 41 patients aged ≤18 years of age, as well as 2 patients with cancer and 1 pregnant patient (Figure 1). A total of 309 abdominal trauma patients with 3242 D-dimer measurements within a 50-day time frame post-trauma were included. Overall, the median age of the included patients was 44.00 (IQR: 31.00–54.00) years, and 250 (80.91%) patients were male. The median SOFA score was 4.00 (IQR: 2.00–7.00), the median AIS of the abdomen was 3.00 (IQR: 3.00–4.00), the median ISS was 20.00 (IQR: 16.00–29.00), and the median TRISS Ps was 0.97 (IQR: 0.92–0.99). The main extra-abdominal trauma was chest trauma (62.78%), and the median AIS of the chest was 2.00 (IQR: 0.00–3.00). The severity of extra-abdominal trauma was relatively minor. There existed a statistically-significant weak correlation between the AIS of the abdomen and AIS of the extremities (with the Spearman’s rank correlation coefficient = 0.13, and *p* = 0.023), while there was no statistically-significant correlation between the AIS of the abdomen and other AIS. Almost all the patients (n = 281, 90.94%) received UFH, and 101 (32.69%) patients received LMWH. Half of the patients (n = 164, 53.07 %) received blood products of plasma, and 55 (17.80%) received blood products of cryoprecipitate. The most frequent complication was liver dysfunction, which presented in 218 (70.55%) patients, while 83 (26.86%) patients developed renal dysfunction. The length of the hospital stay was 18.20 (IQR: 11.59–30.03) days, and the in-hospital all-cause mortality was 7.44% (n = 23). The injury severity, blood transfusion, complications of sepsis, intra-abdominal infection, renal dysfunction, and liver dysfunction were statistically different between the survivors and non-survivors (see Appendix A).

### 3.2. Group Characteristics

After the GBTM analysis and a two-stage model selection process, we identified four trajectories of daily maximum D-dimer measurements, including a combination of two quadratic and two quartic polynomials (Figure 2). The model entropy was 0.93, and the APPA of each group was high (i.e., 0.98, 0.91, 0.98, and 1.00). The temporal patterns of the daily D-dimer measurements, clinical characteristics, and in-hospital all-cause mortality differed among the four trajectory groups.

#### 3.2.1. D-Dimer Trajectory Characteristics

Distinct patterns of the daily maximum D-dimer measurements post-trauma were reflected in these four trajectory groups (Figure 2). They presented more clinical features for trauma patients compared to the single spline trajectory (see the Individual trajectories and spline curve and Appendix A). Group 1 (n = 178, 57.61%; “stable low”) was characterized by persistently low D-dimer measurements over time with a mean D-dimer level of 1.39 (IQR: 0.69–3.00) mg/L. Group 2 (n = 87, 28.16%; “moderate-decline”) had a moderate D-dimer measurement on day 0, which gradually decreased over time. Group 3 (n = 26, 8.41%; “high-rapid decline”) demonstrated an extremely high measurement on day 0, a significant declination during the first five days post-trauma, and a slight rise later. Group 4 (n = 18, 5.83%; “high-gradual decline”) was characterized by an extremely high D-dimer on day 1, followed by a gradual decrease from day 1 to day 25. The patients in group 4 had the higher maximum, mean and medium D-dimer measurements than the patients in groups 1 and 2 (Table 1).

#### 3.2.2. Clinical Characteristics

Comparisons of the clinical characteristics between the groups were summarized in Table 1. The ISS (*p* = 0.001) and SOFA score (*p* = 0.005) were higher in groups 3 and 4 than in groups 1 and 2. Almost half of the patients had an AIS of the abdomen >3 for each group, and no difference of the AIS of the abdomen was observed among the groups (see Appendix A). The TRISS Ps was lower in group 4 than in the other groups (*p* = 0.001). The proportion of head trauma was higher in group 3 (n = 14, 53.85%) than in the other groups (*p* = 0.013), while trauma of the face, chest, extremities, and external were not different among the groups. There were statistically-significant differences in the AIS of the head, AIS of the chest, AIS of the extremities, and AIS of the external among the four groups. The median age was highest in group 4 (50.50 years of age), but no statistical differences were observed among the four groups. Clinical treatments of LMWH (*p* < 0.001), blood transfusion (*p* = 0.009), and a transfusion of plasma (0.007) were statistically different among the four groups. No difference in the use of LMWH was observed between groups 3 and 4 (*p* > 0.999). There was no difference across the trajectory groups in the use of UFH and a transfusion of cryoprecipitate. Complications of renal dysfunction were higher in groups 3 and 4 compared to groups 1 and 2 (*p* < 0.001). Complications of VTE, sepsis, intra-abdominal infections, and liver dysfunction were not different among the groups. The length of hospital stay of group 4 was higher than in all the other groups (*p* = 0.001).

### 3.3. Association of D-Dimer Trajectories and In-Hospital All-Cause Mortality

The in-hospital all-cause mortality was distinctively different among the trajectory groups (*p* = 0.004). The mortality was 4.49% in group 1, 9.20% in group 2, 7.69% in group 3, and 27.78% in group 4. Using group 1 as a reference, the patients in group 4 had an increased odds ratio (OR) of in-hospital all-cause mortality (OR = 6.94, 95% CI: 1.20–40.25) after an adjustment for the BMI, SOFA, and AIS of the abdomen. Patients in group 2 (OR = 2.83, 95% CI: 0.63–12.69) and group 3 (OR = 3.11, 95% CI: 0.46–21.24) did not increase their risk of mortality (Table 2). The estimated effects were compatible in the sensitivity analysis after an adjustment of the BMI, SOFA, and ISS, although they did not reach statistical significance (see Appendix A). The AUC was higher in the D-dimer trajectory model than in the TRISS (i.e., 0.957 versus 0.829) (see Appendix A).

## 4. Discussion

Our study used a trajectory modeling approach (GBTM) to investigate the long-term patterns of dynamic D-dimer in abdominal trauma. We identified four distinct trajectory groups of daily maximum D-dimer measurements: the stable low, “moderate-decline”, “high-rapid decline”, and “high-gradual decline” groups. Among the four groups, the D-dimer trajectory presented in the “high-gradual decline” group was an independent predictor of the in-hospital all-cause mortality. No LMWH or UFH differences were presented between the “high-rapid decline” group and the “high-gradual decline” group.

Compared with the static assessment of D-dimer, our study underscores the value of monitoring and evaluating the trajectory patterns of D-dimer in trauma patients. Previous studies evaluating D-dimer observed variations in the cut-off values. For example, Lee et al. found that patients with a D-dimer higher than 34.53 mg/L measured in the first 24 h after trauma had a 1.033-fold (95% CI: 1.016–1.051) higher risk of 28 day mortality, while Ishii et al. showed that D-dimer measured at admission as being higher than 110 mg/L was significantly associated with the 28-day mortality (OR = 5.89, 95% CI: 2.78–12.70) [10,30]. One reason for these inconsistent results is that they assessed the effect of D-dimer based on the single values of D-dimer measured at different time points post-trauma. D-dimer tested at a single time point post-trauma may introduce bias due to the different time windows of disease. In a study of longitudinal D-dimer, Jiang et al. found that patients had probabilities to present both increasing and decreasing D-Dimer levels in the first 24 h post-trauma [31]. In this study, transient high D-dimer measurements were widespread in the early phase in groups 3 and 4; however, only patients in group 4 presented a slowly resolved D-dimer and had a statistically-higher risk of mortality in reference to group 1. The D-dimer trajectory in group 3, which was rapidly alleviated, led to favorable health outcomes. A single-time D-dimer measurement could not sufficiently present a comprehensive picture of the coagulation changes during hospitalization.

Variations in the clinical characteristics and dynamic patterns by D-dimer trajectory memberships can be used to understand the complex mechanisms of coagulation progression in trauma patients. For example, compared with groups 1 and 2, the patients in groups 3 and 4 presented higher D-dimer measurements during long-term post-trauma. These patients had a higher SOFA, ISS, and morbidity of renal dysfunction, suggesting that coagulation dysfunction is associated with injury severity and organ dysfunction. First, pathophysiological research has found that tissue factor released by injured tissue triggers the extrinsic coagulation pathway and promotes the generation of thrombin and fibrin, resulting in the elevation of D-dimer [32]. Severely injured patients may have slow-to-heal injuries contributing to an ongoing activation of coagulation. Moreover, thrombi generated from coagulation dysfunction occludes the microvascular circulation of different organs, resulting in organ dysfunction. Moreover, sustained organ failure further accelerates clotting [33].

Temporal patterns of D-dimer within five days post-trauma can be used to characterize patients into group 3 or group 4 and, thus, could be helpful in guiding therapy. Recent studies have underlined the heterogenous course of trauma, and subgroups of patients could benefit from targeted support [34,35]. In this study, we found that patients in groups 3 and 4 had similar clinical characteristics, severity of trauma, and usage of UFH and LMWH, but they showed different progressions of D-dimer and different risks of mortality. Whether a patient with initially high D-dimer will move to group 4 is uncertain at the time of trauma, and D-dimer monitoring is especially beneficial for trauma patients with initially-high D-dimer. Moreover, the different patterns of D-dimer between groups 3 and 4 suggest varying dynamic treatment-responsive pathophysiology in abdominal trauma patients. Initially-high followed by slowly-descending D-dimer in the first five days post-trauma might function as the marker of disease deterioration. In clinical practice, if patients have such D-dimer patterns after treatment, then clinicians might have to change the treatment strategy to improve the prognosis of trauma. Although UFH and LMWH are the most commonly used anticoagulants and have been shown to be beneficial in trauma patients [6], other subgroup-specific treatment is needed for patients with early-high and slowly-resolved D-dimer.

As a widely used trajectory analysis in clinical research, GBTM provides us with a new tool for statical visualization of the D-dimer progression in hospitalized abdominal trauma patients [36]. Compared to other trajectory analyses, such as growth mixture modeling, GBTM assumes no trajectory heterogeneity within subgroups [37]. This method was suitable for this study to explore how many patterns of longitudinal D-dimer existed in the abdominal trauma patients and it provided opportunities to explore the differences between the trajectory groups, identify the trajectories leading to damage outcomes, and ultimately to optimize the treatment strategies for specific subgroups. In addition, unlike the previous study that compared the D-dimer trajectory between patients with and without VTE [17], GBTM enables us to perform a post hoc analysis to explore more different patterns of D-dimer.

Our study had several limitations. First, as a retrospective study, the D-dimer measurements were collected as clinically recorded, resulting in an inconsistent data density. In order to model non-linear trajectories, 227 patients were excluded due to the limited frequency of D-dimer measurements. This exclusion could have introduced selection bias as severe patients were more likely to be included (see Appendix A). Second, some patients may have been transferred from other hospitals. We did not have data about the patients’ clinical performances and treatments in other hospitals, and we had little information about the D-dimer measurements on day 1 post-trauma for patients in group 4. Although we excluded patients with trauma-hospitalization intervals longer than two weeks to reduce the information gap, there existed an information bias. Furthermore, the sample size limited meaningful statistical comparisons of mortality among the groups after confounding adjustments. Finally, a single-center study and a high proportion of males limited the generalizability to other locations of care and female patients.

## 5. Conclusions

In conclusion, this study introduced a new insight to understand the heterogeneity of D-dimer progression patterns and their associations with mortality by combining GBTM and logistic regression. Four distinct patterns of the changes of D-dimer post-trauma were identified. Patients in the “high-gradual decline” group had an increased risk of in-hospital all-cause mortality, and early monitoring of D-dimer is useful for subtype identification and therapy improvement. This study set the foundational groundwork for patient-centered coagulation management pathways. A larger prospective study is needed to examine the other potential differences between D-dimer groups to enhance understanding of the physiological process and precision medicine of coagulation in trauma.

## Figures and Tables

**Figure 1 jcm-12-01091-f001:**
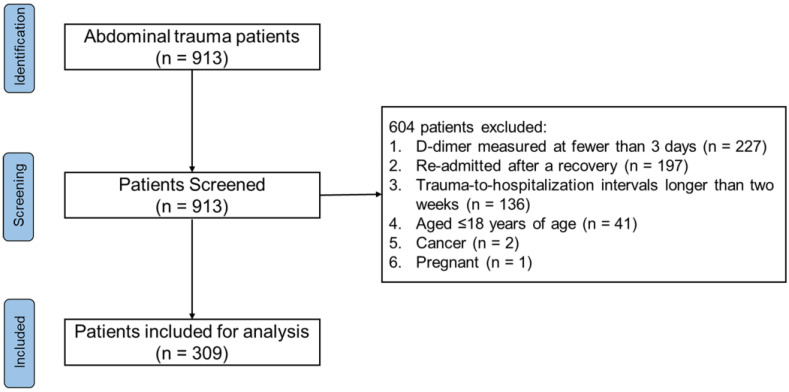
Flowchart of patients.

**Figure 2 jcm-12-01091-f002:**
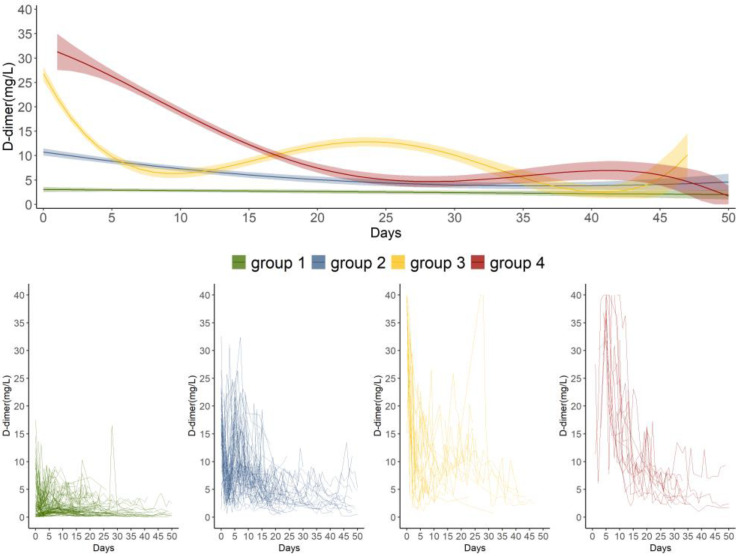
D-dimer trajectories as defined by group-based trajectory modeling over the first 50 days post-trauma. Day 0 was defined as the day of trauma occurrence. Group 1: stable low group (n = 178, 57.61%); Group 2: “moderate-decline” group (n = 87, 28.16%); Group 3: “high-rapid decline” group (n = 26, 8.41%); Group 4: “high-gradual decline” group (n = 18, 5.83%). The top graph presents the mean D-dimer measurement with the 95% CI for the distinct trajectory groups each day. The bottom graphs show the individual trajectories in the four trajectory groups.

**Table 1 jcm-12-01091-t001:** Characteristics among trajectory groups.

	All Patients (n = 309)	Group 1 (n = 178)	Group 2 (n = 87)	Group 3 (n = 26)	Group 4 (n = 18)	*p*
**Baseline Characteristics**						
Age, yr, median (IQR)	44.00 (31.00, 54.00)	42.00 (31.00, 52.75)	44.00 (31.00, 54.50)	48.50 (34.25, 58.00)	50.50 (36.50, 59.50)	0.173
Male gender, n (%)	250 (80.91)	145 (81.46)	72 (82.76)	17 (65.38)	16 (88.89)	0.170
BMI, median (IQR)	22.04 (20.66, 24.11)	21.92 (20.70, 23.88)	22.46 (20.36, 25.15)	22.63 (20.83, 23.97)	22.29 (21.15, 27.62)	0.347
**Extra-abdominal trauma**						
Head, n (%)	86 (27.83)	47 (26.40)	19 (21.84)	14 (53.85)	6 (33.33)	0.013
Face, n (%)	9 (2.91)	6 (3.37)	2 (2.30)	0 (0.00)	1 (5.56)	0.689
Chest, n (%)	194 (62.78)	104 (58.43)	59 (67.82)	17 (65.38)	14 (77.78)	0.241
Extremities, n (%)	108 (34.95)	54 (30.34)	32 (36.78)	12 (46.15)	10 (55.56)	0.086
External, n (%)	86 (27.83)	58 (32.58)	19 (21.84)	3 (11.54)	6 (33.33)	0.064
**Severity of Trauma**						
**AIS**						
Head, median (IQR)	0.00 (0.00, 1.00)	0.00 (0.00, 1.00)	0.00 (0.00, 0.00)	1.50 (0.00, 3.00)	0.00 (0.00, 2.75)	0.011
Face, median (IQR)	0.00 (0.00, 0.00)	0.00 (0.00, 0.00)	0.00 (0.00, 0.00)	0.00 (0.00, 0.00)	0.00 (0.00, 0.00)	0.689
Chest, median (IQR)	2.00 (0.00, 3.00)	2.00 (0.00, 3.00)	3.00 (0.00, 3.00)	3.00 (0.00, 3.00)	3.00 (2.25, 3.75)	0.003
Extremities, median (IQR)	0.00 (0.00, 2.00)	0.00 (0.00, 2.00)	0.00 (0.00, 3.00)	0.00 (0.00, 3.00)	3.00 (0.00, 4.00)	0.008
External, median (IQR)	0.00 (0.00, 1.00)	0.00 (0.00, 1.00)	0.00 (0.00, 0.00)	0.00 (0.00, 0.00)	0.00 (0.00, 1.00)	0.049
Abdomen, median (IQR)	3.00 (3.00,4.00)	3.50 (2.00,4.00)	3.00 (3.00,4.00)	3.00 (3.00,4.00)	4.00 (2.25,4.00)	0.964
AIS of the abdomen > 3, n (%)	153 (49.51)	89 (50.00)	43 (49.43)	11 (42.31)	10 (55.56)	0.845
ISS, median (IQR)	20.00 (16.00, 29.00)	18.00 (13.25, 26.00)	24.00 (16.00, 32.00)	25.00 (18.00, 34.00)	34.00 (20.75, 34.75)	0.001
SOFA, median (IQR)	4.00 (2.00, 7.00)	3.00 (2.00, 6.00)	3.00 (2.00, 5.50)	5.50 (4.00, 9.00)	7.50 (3.00, 9.50)	0.005
TRISS Ps, median (IQR)	0.97 (0.92, 0.99)	0.98 (0.94, 0.99)	0.97 (0.92, 0.99)	0.95 (0.87, 0.98)	0.91 (0.80, 0.96)	0.001
**Clinical Treatments**						
UFH, n (%)	281 (90.94)	158 (88.76)	81 (93.10)	24 (92.31)	18 (100.00)	0.338
LMWH, n (%)	101 (32.69)	39 (21.91)	43 (49.43)	11 (42.31)	8 (44.44)	<0.001
Blood transfusion, n (%)	173 (55.99)	92 (51.69)	46 (52.87)	21 (80.77)	14 (77.78)	0.009
Plasma, n (%)	164 (53.07)	89 (50.00)	41 (47.13)	20 (76.92)	14 (77.78)	0.007
Cryoprecipitate, n (%)	55 (17.80)	27 (15.17)	17 (19.54)	7 (26.92)	4 (22.22)	0.433
Surgery, n (%)	200 (64.72)	111 (62.36)	62 (71.26)	18 (69.23)	9 (50.00)	0.261
**Hospital Complications and Outcomes**						
VTE, n (%)	25 (8.09)	9 (5.06)	11 (12.64)	2 (7.69)	3 (16.67)	0.093
Sepsis, n (%)	71 (22.98)	31 (17.42)	26 (29.89)	8 (30.77)	6 (33.33)	0.059
Intra-abdominal infection, n (%)	51 (16.50)	29 (16.29)	16 (18.39)	2 (7.69)	4 (22.22)	0.547
Renal dysfunction, n (%)	83 (26.86)	32 (17.98)	27 (31.03)	15 (57.69)	9 (50.00)	<0.001
Liver dysfunction, n (%)	218 (70.55)	121 (67.98)	61 (70.11)	22 (84.62)	14 (77.78)	0.320
LOS, days, median (IQR)	18.20 (11.59, 30.03)	15.98 (10.90, 25.13)	22.72 (15.05, 35.44)	20.06 (11.62, 33.52)	26.36 (17.15, 41.46)	0.001
In-hospital all-cause death, n (%)	23 (7.44)	8 (4.49)	8 (9.20)	2 (7.69)	5 (27.78)	0.004
**D-dimer Characteristics**						
Maximum D-dimer, mg/L, median (IQR)	7.77 (2.14, 17.80)	2.46 (1.27, 5.83)	16.02 (11.00, 21.26)	34.08 (27.35, 40.00)	33.88 (25.02, 40.00)	<0.001
Mean D-dimer, mg/L, median (IQR)	3.83 (1.25, 7.92)	1.39 (0.69, 3.00)	7.42 (5.96, 9.66)	11.30 (9.76, 13.59)	11.62 (9.93, 18.60)	<0.001
Medium D-dimer, mg/L, median (IQR)	3.29 (1.09, 6.55)	1.32 (0.66, 2.38)	6.59 (5.18, 8.78)	7.89 (5.95, 9.85)	8.42 (7.68, 13.92)	<0.001
Minimum D-dimer, mg/L, median (IQR)	1.41 (0.47, 2.62)	0.58 (0.24, 1.39)	2.61 (1.61, 4.07)	3.13 (2.56, 4.22)	2.70 (2.24, 7.33)	<0.001
SD of D-dimer, mg/L, median (IQR)	2.06 (0.50, 5.07)	0.61 (0.30, 1.50)	4.39 (2.65, 5.91)	9.34 (7.06, 11.60)	9.01 (6.68, 11.84)	<0.001

Group 1: stable low group; Group 2: “moderate-decline” group; Group 3: “high-rapid decline” group; Group 4: “high-gradual decline” group. Definition of abbreviation: IQR: interquartile range; BMI: Body Mass Index; AIS: Abbreviated Injury Scale; ISS: Injury Severity Score; SOFA: Sequential Organ Failure Assessment; TRISS Ps: the probability of survival obtained by the Trauma Injury Severity Score; UFH: unfractionated heparin; LMWH: low-molecular-weight heparin; VTE: venous thromboembolism; LOS: length of hospital stay; SD: standard deviation.

**Table 2 jcm-12-01091-t002:** Association between trajectory groups and in-hospital all-cause mortality.

Trajectories	Univariate Analysis		Multivariate Analysis	
	OR (95% CI)	*p*	OR (95% CI)	*p*
Group 1	Reference		Reference	
Group 2	2.15 (0.78, 5.94)	0.139	2.83 (0.63, 12.69)	0.175
Group 3	1.77 (0.35, 8.84)	0.486	3.11 (0.46, 21.24)	0.247
Group 4	8.17 (2.34, 28.57)	0.001	6.94 (1.20, 40.25)	0.031

Adjusted by BMI, SOFA, and AIS of the abdomen. Although the ISS had a *p*-value of <0.1 in the univariate analysis, it was correlated with the AIS of the abdomen (r = 0.52, *p* < 0.001). We further adjusted the BMI, SOFA, and ISS in the sensitivity analysis. Group 1: stable low group; Group 2: “moderate-decline” group; Group 3: “high-rapid decline” group; Group 4: “high-gradual decline” group. OR: odds ratio; CI: confidence interval.

## Data Availability

The data presented in this study are available on request from the corresponding author. The data are not publicly available due to patient privacy concerns.

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
