# Peer review of "Longitudinal D-Dimer Trajectories and the Risk of Mortality in Abdominal Trauma Patients: A Group-Based Trajectory Modeling Analysis"

_jcm, 2023, doi:10.3390/jcm12031091_

Round 1

Reviewer 1 Report

*  The opening paragraph in the Introduction is a bit fragmented.  You mention that coagulation dysfunction is a notable feature (usually hypocoagulation), yet you state abruptly that these persons are generally placed on low dose anticoagulants, presumably for venous thromboembolism prophylaxis since such patients are at risk for this too.  Shouldn't you state this to make the condition clear?

*  The problem with D-dimer is that it is TOO sensitive (lots of false positives for clinically important events), so does not have great clinical utility.  However, looking at the dynamics rather than static values might yield better results, and that is your hypothesis.  Seems reasonable and should be stated as such.

*  Why was this limited only to abdominal trauma?  What about traumatic brain injury?  Extremity fractures?

*  For the d-dimer measurements, it seems you took convenience clinical data.  What is the norm for testing and frequency in your centre?  This will provide some context for the reader.

*  Seems like you should also collect use of other blood products than just blood transfusions.  What about plasma?  Cryoprecipitate?

*  Why were those with hospitalisation times longer than 2 weeks excluded?

*  For the statistical analysis, it appears that the injury severity (ISS) is much higher for groups 3/4.  Was this controlled in the regressions?  Most of the time mortality is tightly associated with injury severity to the exclusion of almost all other variables.

*   I like the way you've handled the data in particular the categorisation of the populations which are appropriately disparate.  However, injury severity must be considered closely in the categorisation.  The real marginal risk of d-dimer within a particular category it seems is what is most important.  In that vein, you show it is only discriminative in group 4, which has the most injury severity by a long shot.  Does this mean that d-dimer measurement is only beneficial in this group?  You mention this some in the Discussion, but it should be emphasised more. 

Reviewer 2 Report

Introduction

What could be the real impact on define trajectories based on DD ? It is posible to change the treatment?

Methods

The are too much patients excluded (from 913 to 309), a PRISMA figure could improve the report of exclusions?

Authors should clarify how did define abdominal trauma?

Did they exclude patients with adquired trombosis during admission.?

 Minimun AIS to be included? They should describe the distribution of of included patients across the AIS spectrum.

There are others scores more accurate to use in trauma pop (like TRISS )

There several tools for score prediction in trauma. Authors must compare the propose trajectories with these tools (TRISS, lactate or EB at admission etc). 
